# Incorporating Patient Preferences into a Decision-Making Model of Hand Trauma Reconstruction

**DOI:** 10.3390/ijerph182111081

**Published:** 2021-10-21

**Authors:** Dun-Hao Chang, Yu-Hsiang Wang, Chi-Ying Hsieh, Che-Wei Chang, Ke-Chung Chang, Yo-Shen Chen

**Affiliations:** 1Division of Plastic and Aesthetic Surgery, Department of Surgery, Far Eastern Memorial Hospital, No. 21, Section 2, Nanya S. Road, Banciao District, New Taipei City 22060, Taiwan; jchang50002000@gmail.com (D.-H.C.); hchiwen@hotmail.com (C.-Y.H.); b92401068@gmail.com (C.-W.C.); chang661010@yahoo.com.tw (K.-C.C.); 2Department of Information Management, Yuan Ze University, No. 135 Yuan-Tung Road, Chung-Li, Taoyuan City 32003, Taiwan; 3School of Medicine, National Yang Ming Chiao Tung University, No. 155, Section 2, Linong Street, Taipei 11221, Taiwan; 4Department of Surgery, Chung Shan Medical University Hospital, No. 110, Section 1, Jianguo North Road, Taichung City 40201, Taiwan; 99311132tzu@gmail.com; 5Graduate Institute of Biomedical Electronics and Bioinformatics, National Taiwan University, No. 1, Section 4, Roosevelt Road, Taipei 10617, Taiwan; 6School of Medicine, National Taiwan University, No. 1, Section 4, Roosevelt Road, Taipei 10617, Taiwan

**Keywords:** conjoint analysis, patient preference, hand reconstruction, decision-making

## Abstract

Background: Few studies have addressed patient preferences in emergent surgical decision making. Aim of the study: Analyzing patient preferences for hand trauma reconstruction to propose a decision-making model. Methods: A conjoint analysis survey was developed with Sawtooth Software. Three common flaps—i.e., a cross-finger flap (CFF), a dorsal metacarpal artery perforator flap (DMAPF), and an arterialized venous flap (AVF)—were listed as treatment alternatives. Five attributes corresponding to these flaps were recovery time, total procedure, postoperative care methods, postoperative scar condition, and complication rate. Utility and importance scores were generated from the software, and preference characteristics were evaluated using cluster analysis. Results: The survey was completed by 197 participants with hand trauma. Complication risk received the highest importance score (42.87%), followed by scar condition (21.55%). Cluster analysis classified the participants as “conservative,” “practical,” and “dual-concern”. The dual-concern and conservative groups had more foreign laborers and highly educated participants, respectively, than the other groups. Most participants in the conservative and practical groups preferred DMAPF, whereas those in the dual-concern group favored CFF. Our proposed model consisted of shared decision-making and treatment recommendation pathways. Conclusion: Incorporating patient preferences into the decision-making model can strengthen patient-centered care. Further research on the applications of the proposed model is warranted.

## 1. Introduction

Surgical decision making has historically been based on existing guidelines, personal experiences, and recommendations with the application of surgical basic sciences. However, many factors may affect decisions, including clinical states, healthcare resources, surgeons’ expertise, and patient preferences [1].

Currently, patient-centered care has become a central value in healthcare systems, and this represents a shift in the traditional roles of patients and their families, from only taking orders to becoming team participants. Therefore, the trend of medical decision making has changed from passive informed consent to shared decision making (SDM). A fundamental aspect of SDM involves having an understanding of patient preference and how patients weigh benefits and risks when considering different treatment alternatives [2].

To accurately measure patient preferences, there is growing interest in the use of conjoint analysis (CA) [3]. CA, initially developed and used in market analysis, is a method designed to elicit preferences through surveys. CA assesses the willingness to accept trade-offs between attributes and provides estimates of the most important factors to patients when deciding between treatments. There have been many studies using CA to explore patients’ perspectives in medical decision making, such as medication initiation and elective surgery [4,5,6].

However, there is a scarcity of studies addressing patient preferences in emergent surgical decision making. In an emergent setting, owing to limited time and the unavailability of complete information, surgeons typically would have made decisions on the basis of habitual practice, and patients’ preferences have seldom been considered, which may have led to decision regret. Such situations have arisen in daily clinical practice, particularly in the case of trauma patients.

Therefore, in this study, we created a common scenario of hand trauma and used a CA tool to weigh utility and importance among the attributes in surgical decision making. Based on the study results, we aim to incorporate the findings into the proposed decision-making model for future clinical application.

## 2. Materials and Methods

The study was approved by the Institutional Review Board of Far Eastern Memorial Hospital. Owing to the fact that real-world decision making pertaining to the treatment of patients with hand trauma usually occurs in the emergency room (ER), conducting a survey is often difficult in such a chaotic environment. Thus, the study population evaluated in the present study comprised patients who had experienced hand trauma (various types of hand injuries) and underwent treatment in the ER or operating room. They were recruited to complete the survey during clinic visits. The participants had to be at least 20 years of age. Some foreign workers were also included by providing a special-edition survey according to their languages. All surveys were completed with the aid of an instructor who helped the participants to understand the details of the statements. Each participant could obtain NTD 100 (USD 3.5) after completion as an incentive.

### 2.1. CA: Attribute and Value Level Determination

We set the scenario of the survey as follows: “You have had an injury causing a soft tissue defect at the little finger” (Figure 1). Then, the respondents had to provide their level of acceptance according to different attributes and further choice-based conjoint (CBC) tasks.

To identify potential attributes, we conducted a literature review surveying surgical details, complications, and outcomes of the three common flaps of choice, i.e., cross-finger flap (CFF), dorsal metacarpal artery perforator flap (DMAPF), and arterialized venous flap (AVF). 

CFF is applied as a two-stage reconstruction. It is elevated from the adjacent finger and transposed to cover the defect, and the secondary defect is closed by a skin graft. The fingers are immobilized for approximately 2–3 weeks, after which a secondary procedure is performed to divide the skin bridge [7]. DMAPF is a perforator-based flap that is harvested from the dorsal hand. It can be achieved through one-stage reconstruction but leaves an unsightly scar at the back of the hand. At times, venous congestion occurs postoperatively because of its circulation direction, thereby leading to partial flap necrosis [8]. AVF is a type of free flap that has arterial inflow through an afferent vein perfusing the flap and subsequent venous outflow through the efferent veins. The usual donor sites include the volar forearm, medial arm, and dorsal foot. A microsurgical technique is required to perform vascular anastomosis; therefore, complications, such as venous congestion, ischemia, and loss of flaps, are common [9].

These three flaps are thin and pliable and suitable for finger reconstruction. Focusing on skin coverage, we did not include postoperative hand function (e.g., range of motion, sensation, and stiffness), which is usually more related to fracture, tendon, or nerve injury. Regarding payment, Taiwan launched a national healthcare program, National Health Insurance (NHI), in 1995. The NHI has covered >99% of the population, including foreign workers, and out-of-pocket payments are similar among the flaps. Thus, payment was not taken into consideration as a candidate attribute. The attribute selection process was discussed thoroughly among our hand surgery team, consisting of three specialists who had 35 years, 8 years, and 5 years of experience in performing hand surgeries.

Five attributes had been selected in the final version of the survey, namely, recovery time, total procedures, postoperative care methods, postoperative scar condition, and complication rate. Each attribute had 2–3 value levels incorporating findings from the literature review. Recovery times were set as the usual interval of stitch removal, which were 2 and 4 weeks for one- and two-stage surgeries, respectively. The scar conditions were described for the donor sites of the flaps, and we selected the most common site as a representative, for example, groin area for full-thickness skin graft in a CFF and medial forearm for AVF. Complication rates were defined according to complications that required reoperation, such as flap ischemia and necrosis; thus, we set complication rate 1% for CFF, 5% for DMAPF, and 10% for AVF, the most difficult [8,9,10]. The details are presented in Table 1.

This set of attributes and value levels generated 72 different (2 × 3 × 2 × 3 × 3) hypothetical flaps of choice, forming 2556 groups of two comparisons. Given the massive number of choices produced by this model, testing all of them was not practical. Thus, we used Sawtooth Software to conduct adaptive discrete choice experiments (DCEs). Sawtooth Software could generate random pairs of choices according to its algorithm, which is orthogonal, making the participants see an equal number of times of attribute pairs. Moreover, this tool had been used in several studies and had proven its feasibility and reliability [6,11].

The respondents needed to select the most desirable surgery between hypothetical surgeries A and B, each of which consisted of five attributes with differing value level combinations. Figure 2 shows a sample DCE. Most attributes and levels were explained in plain language, except for the scar condition, which was illustrated in a schematic diagram for better understanding.

Sawtooth Software uses the empirical Bayes method to calculate a logit solution using all DCE data to generate aggregate utilities. Utilities represent the relative desirability of each value level with the attributes, also known as part worth. Utility scores are anchored to 0; higher positive and negative scores indicate that the participants have relatively higher and lower desirability, respectively, in selecting the options based on the value levels. To further clarify, the raw utilities are normalized and transformed to a scale wherein the average difference between best and worst levels across attributes is equal to 100. From part-worth utilities, we calculated the attribute importance score in which importance = [(utility range)/(sum of utility range for all characteristics)] × 100; this further quantified the relative influence of each attribute on participants’ decision making.

### 2.2. Analysis and Statistics

We determined the sample size according to the rule of thumb proposed by Johnson and Orme, the founders of Sawtooth Software [12]. The rule suggests that the sample size required for the main effects is N > 1000 [(largest number of value levels for any of the attributes)]/[(number of DCE) (number of alternatives in each DCE)]. In our study, the required minimal sample size was [(1000) (3)/(11) (2)] = 136. To allow for a margin of error, we set the target case number as 200.

For the utilities and importance scores, we used mean and standard deviation to demonstrate a global preference. We further performed a cluster analysis to investigate preference characteristics. Cluster analysis is an unsupervised learning approach of data mining that organizes a set of objects into groups according to their similarities. The first step is to conduct hierarchical clustering on the 12 utilities using Euclidean distance and Ward’s minimum variance methods. We subsequently also performed K-means clustering according to the number of clusters suggested from the first step. The optimal clustering results were verified using a silhouette coefficient, with higher values indicating that the object is well matched to its own cluster. The average utilities and importance scores of each group were evaluated, and the preference was named. The demographics and characteristics of the clusters were also analyzed. 

Treatment preferences among the three proposed flaps (CFF, DMAPF, and AVF) were evaluated using first-choice and share-of-preference models. The first-choice model is also called the maximum utility rule, in which utilities are summed across the levels corresponding to each option (flap), and the one with highest utility is the preferred choice. For example, a patient’s utilities (raw utility) are −44 (−0.77) for two-stage surgery, −14 (−0.25) for 4 weeks recovery, −40 (−0.70) for post-op finger connection care, 2 (0.03) for adjacent finger + groin scar, and 130 (2.22) for 1% complication risk; and the aggregated utility of CFF would be [(−44) +(−14) + (−40) + (2) + (130)] = 34(0.54). If the sum utility of DMAPF is 132 (1.68) and AVF is −67 (−0.51), the first-choice model would determine that the patient chooses DMAPF. The share-of-preference model is based on logit probability, using the exponentiated utility (expU) of a certain option divided by the sum of expU of all options. However, the raw utility (not transformed) is used for this simulation. With the same example, the patient’s probability of choosing DMAPF is exp (1.68) / [exp (1.68) + exp (0.54) + exp (−0.51)] = 69.8%, 22.3% for CFF, and 7.8% for AVF.

We used analysis of variance and χ 2 tests for continuous and categorical variables, respectively, with SPSS software for Windows (version 25, SPSS, Chicago, IL, USA). A *p*-value <0.05 was considered significant.

## 3. Results

### 3.1. Characteristics of Participants

Between 15 August 2020 and 30 April 2021, 200 participants were enrolled, and a total of 197 participants completed the survey, while 3 had some missing information. Among the 143 men and 54 women (mean age, 44.1 years), 18 patients were from foreign countries, including 15 Vietnamese and 2 Burmese nationals, and 1 Indonesian national. These patients had various hand injuries, and 39 (19.8%) of them had skin defects requiring skin graft or flap surgery, which was similar to our assumptive scenario. The demographics and characteristics of participants are presented in Table 2.

### 3.2. Preference Level Evaluation

Utility among the attributes and values is shown in Table 3. The complication rate of 1% had the highest utility (87.28), whereas 10% had the lowest utility (−104.97). The importance score (Figure 3) demonstrated that complication risk was considered as the most important attribute in decision making, followed by scar condition. However, this was only the global preference. When observing the wide variety of utility levels among the participants (Figure 4), it was noted that not all participants see things in the same way.

### 3.3. Cluster Analysis

The hierarchical clustering with all utilities grossly grouped the participants into three clusters (Figure A1). We then used k = 3 for the K-means partitioning method for clustering. Comparing these two methods, we selected K-means as the final version of clustering because it had a higher silhouette coefficient than that of hierarchical clustering (0.181 vs. 0.155).

The utility and importance scores according to the clusters are shown in Figure 5. Cluster 1 had the largest population (51%) and was extremely risk averse and accepted two-stage surgery or postoperative inconvenience. Cluster 2 consisted of 23% of participants, who were least worried about complications but were relatively concerned about the operative stages and postoperative care. Cluster 3 accounted for 26% of participants concerned about both scar and complication issues. Framing the preference characteristics of the three clusters, we denoted cluster 1 as “conservative”, cluster 2 as “practical”, and cluster 3 as “dual-concern”. The demographics of the clusters are presented in Table 4. Age and sex distributions were similar in the three clusters, but there were more highly educated participants in cluster 1 and more foreign laborers in cluster 3. Cluster 3 seemed to have more blue-collar patients, while cluster 2 had more patients with actual experiences of skin grafts or flaps. However, the statistical differences were not significant.

As for treatment preferences among the three common flaps, the results are shown in Figure 6. Most patients in cluster 3 preferred CFF, while those in clusters 1 and 2 favored DMAPF. AVF was the least preferred flap in every cluster. The results of treatment preference were similar in the two different methods.

### 3.4. Proposed Decision-Making Model

Based on the study results, we incorporated patient preferences into the decision-making model of hand trauma illustrated in Figure 7. The surgical treatment for hand trauma, or specifically for soft tissue reconstruction, should be related to surgical techniques and hospital resources, patient conditions, and injury types. Considering the aforementioned factors, the authors divided the decision making into two pathways. If time, space, or language allowed for good communication, SDM would be applied. Decision aids were made following the format we used in the CA, clearly describing the complication risks and showing the scarring condition with figures. All the treatment choices would be presented to the patients, and they would finally choose the preferred one. 

However, if communication between doctors and patients was difficult, the treatment recommendation pathway would be suggested. According to the patients’ characteristics, such as age, education, occupation, and nationality, their preference phenotype could be assessed, and suitable treatment (maybe narrowed down to one or two choices) would be suggested. In both pathways, the process of informed consent should be completed. 

## 4. Discussion

In the decision-making process for choosing healthcare treatments, the trade-offs of risks and benefits are particularly important owing to their significant impact on patient quality of life [13]. Understanding patient preferences for treatment options and considering which treatment factors are valued by patients may help in moving decision-making toward patient-centered care [14].

Conjoint analysis using the iterative case-based choices method can disclose the relative importance and ranking that patients place on various attributes of treatment options. This information can then be used as a reference to help both patients and doctors determine what patients are willing to sacrifice to reach treatment goals. Some studies have used CA as a primary decision-making tool and a core element of SDM, which was shown to improve patient satisfaction and decrease decision conflict [6,11].

Our study used CA to examine which factors are most important to patients who will undergo hand soft tissue reconstruction. We found that complication risk had the greatest influence on flap selection, followed by scar condition. The utility differences between 1% and 5% or 5% and 10% complication risks were also significant, which could affect the patients’ final decisions. However, the complication risk (if <10%) is sometimes ignored by surgeons who may assume that the difference is insignificant for patients. For example, the surgeon may recommend a free or perforator-based flap rather than a regional or local flap (traditional flaps) to achieve one-stage reconstruction or a concealed donor site scar, even if the flap necrosis rate is 5% for the former and 1% for the latter. This situation is common in the modern era because of popularity in social media; for example, in the International Microsurgery Club on Facebook [15], surgeons tended to perform or recommend technique-demanding flaps to obtain more “likes”.

However, the success or complication rate of the same surgery may be different among hospitals or surgeons. For example, the finger replantation success rate is related to the volume of practice, which is highly variable [16]. Therefore, since a patient’s decision making is determined by complication rates, a request for a transfer to seek expertise would be suggested if a current service could not fulfill a patient’s expectation.

The other interesting finding in our study is the wide variety of preferences (utility) of the attributes among patients. This finding was quite different from those of previous studies using CA in hand surgery. The possible reason may be that our study population consisted of ER visitors who came from different backgrounds, unlike that of the Harris study on finger joint osteoarthritis, which predominantly consisted of elderly, retired patients [5]. Therefore, it was worthwhile performing clustering to characterize preferences.

A common further step of CA is to classify the preference characteristics of customers or patients, because this allows for the most suitable products or treatments for them to be designed or recommended. Fraenkel et al. have published a serial study addressing medication choices with CA in rheumatoid arthritis [17,18]. The authors used the term “preference phenotype” to categorize various patient preferences. According to the preference phenotype, they could provide patients with an individual treatment plan, and they consequently achieved the goal of “treat to target”.

In our cluster analysis, the three groups had different characteristics and profiles. Their treatment preferences among the three actual flaps would also be distinct, indicating that there is no gold standard for flap reconstruction in terms of patient perspective in finger soft tissue reconstruction. For example, a CFF may be more recommended to the highly educated, white-collar patients, whereas a DMAPF could be considered more suitable for blue-collar foreign workers. Exploration of individual preference and value is important in medical decision making, although this is sometimes ignored in the emergent scenario. Our study demonstrated the importance of patient preferences and how these affected the subsequent decisions.

Regarding attribute selection in our study, some excluded attributes can still be discussed. Although the cost of surgery would be covered by the NHI in Taiwan, some medical products not covered by insurance, such as artificial dermis, can be used as treatment alternatives. However, comparing the functional outcome between flaps and artificial dermis is difficult; thus, we abandoned this treatment choice and the attribute of out-of-pocket cost. Other factors, such as postoperative pain, length of hospital stay, anesthesia methods, and risks, may also play a role in decision making. However, increasing the number of attributes includes increasing the number of CBC tasks that need to be performed. This would consequently increase the complexity and duration of the survey, and participants may lose their patience and concentration. Therefore, our study only provided a profile of patient preferences rather than the full picture. Moreover, there are still many available flaps of choice that we did not include in this study, such as an adipofascial digital artery perforator flap, dorsal digital island flaps, or an abdominal flap [19,20]. Thus, the selected attributes should be adjusted to different applications or national healthcare policies.

Although patient preferences play an important role in SDM, it is difficult to conduct the process in the ER, especially during the COVID-19 pandemic. Strict protection and isolation policies limit communication and discussion between patients and doctors. In this situation, a preference-based treatment recommendation can replace SDM to help patients choose one from many alternative options. Shaoibi et al. developed a Bayesian collaborative filtering algorithm that combines pretreatment preferences and patient-reported outcomes to provide treatment recommendations [21]. In this study, although we used an assumptive scenario and did not include treatment outcome to adjust the recommendation, the proposed decision-making model can act as a prototype for future applications to reduce uncertainty and develop reasonable expectations of the outcome.

Our study still has some limitations. First, most participants (80.2%) in our study did not have the same real finger soft tissue defects as the scenario in the survey. Therefore, the hypothetical question may not reflect their real thinking process. Second, there is inconsistency in the CA. In our study, except for the scar condition, other attributes had obvious superiority of value levels. However, we found that some participants had the opposite utility result (e.g., 4 weeks > 2 weeks recovery time). Johnson and Mathews have described that the inconsistent phenomenon is common in CA, and it was noted in 9.8–26% of participants, especially when they focused on only one or two attributes [22]. As the complication risk accounted for the single greatest factor of decision making in our study, certain participants may select the favored treatment by heuristics and violate the theoretical consistency. Simple deletion of these samples may be inappropriate, because these preferences should also be included in social value estimates. However, more comprehensive survey design or better selection of attributes may alleviate this problem.

## 5. Conclusions

To the best of our knowledge, this is the first study to examine medical decision-making in emergency surgery with CA. Complication risk is the most important factor when choosing the flap for hand reconstruction. Cluster analysis is valid in classifying preference characteristics among the various values. Incorporating consideration of patient preferences into the proposed decision-making model can strengthen patient-centered care. Further research into the applications of the decision model is warranted.

## Figures and Tables

**Figure 1 ijerph-18-11081-f001:**
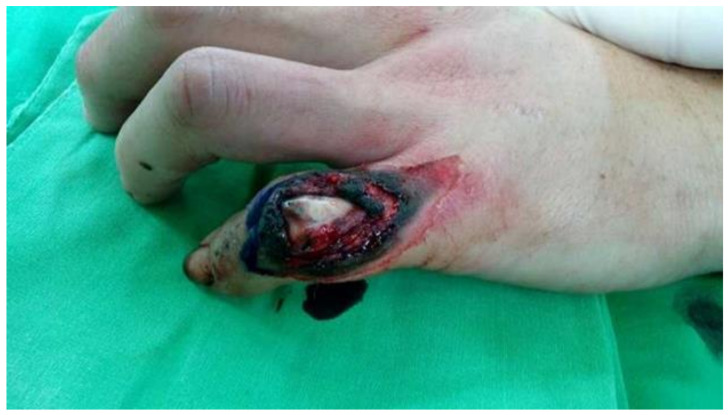
The assumptive scenario in the survey.

**Figure 2 ijerph-18-11081-f002:**
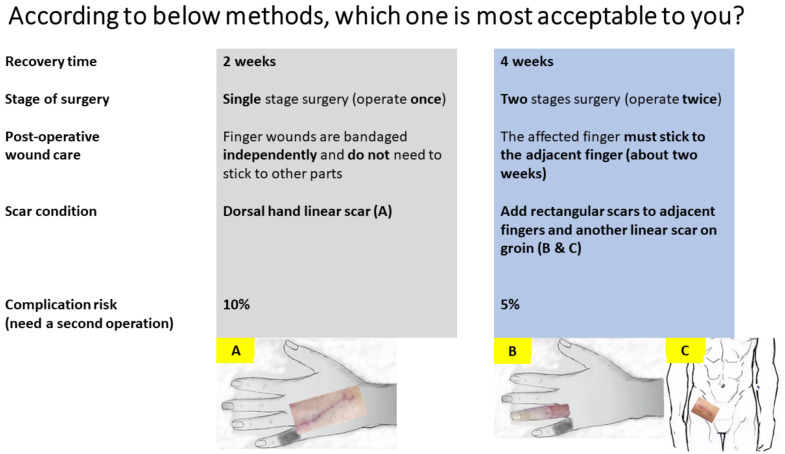
Representative example of 1 of 11 hypothetical choice experiments in the survey. The participants were asked to select one from the two treatment options. (**A**) Dorsal hand scar; (**B**) Finger scar; (**C**) Groin scar.

**Figure 3 ijerph-18-11081-f003:**
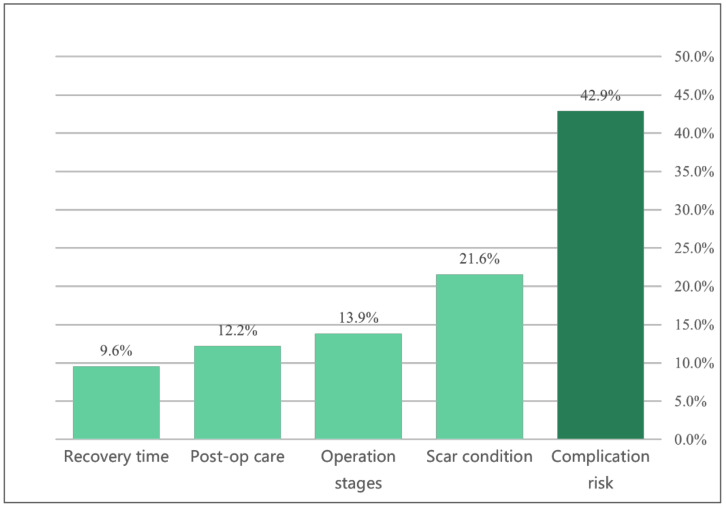
Relative importance of the five attributes of finger reconstruction surgery, scaled to the sum of 100%. The values listed are representative of the average values across participants.

**Figure 4 ijerph-18-11081-f004:**
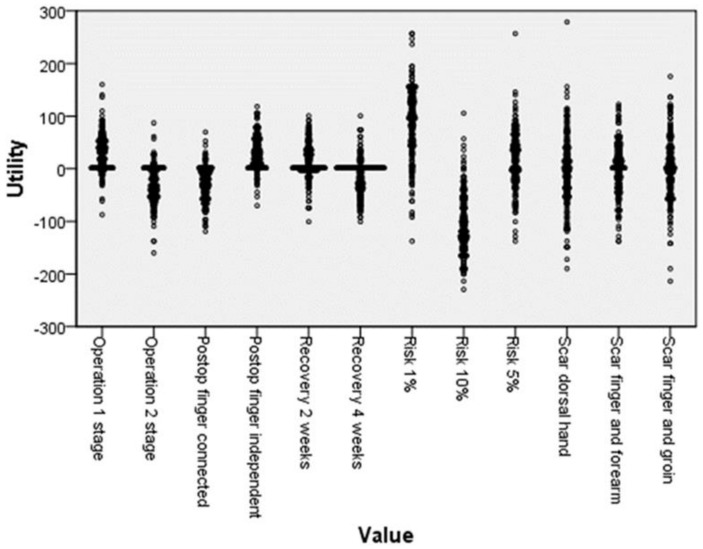
Distribution of the utility level, showing a wide variety of choices among participants.

**Figure 5 ijerph-18-11081-f005:**
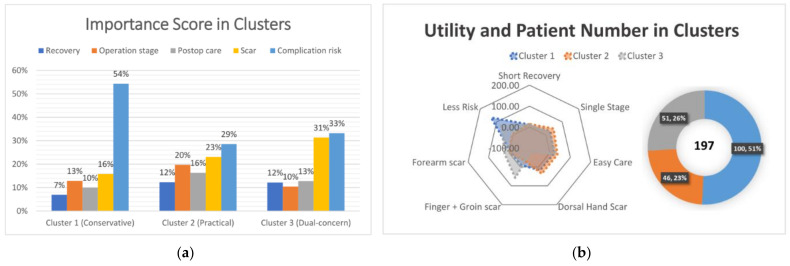
(**a**) Mean utility and patient numbers of the three clusters. The radar plot reveals a striking pattern of differences between clusters. For example, cluster 1 shows a strong preference for less risk compared to other clusters. (**b**) The importance score per cluster.

**Figure 6 ijerph-18-11081-f006:**
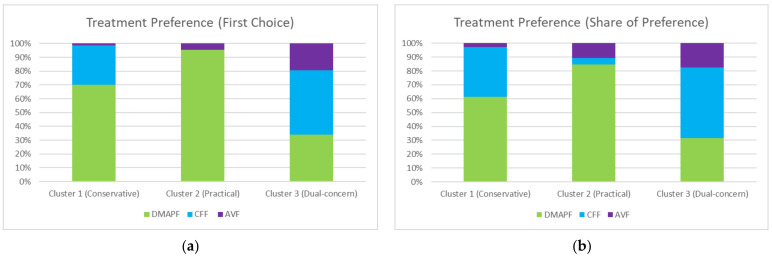
Treatment preferences for the three common flaps: DMAPF (green), dorsal metacarpal artery perforator flap; CFF (blue), cross-finger flap (with groin skin graft); AVF (purple), arterialized venous flap. Most patients in cluster 3 preferred CFF, while clusters 1 and 2 favored DMAPF. (**a**) First-choice method; (**b**) share-of-preference method.

**Figure 7 ijerph-18-11081-f007:**
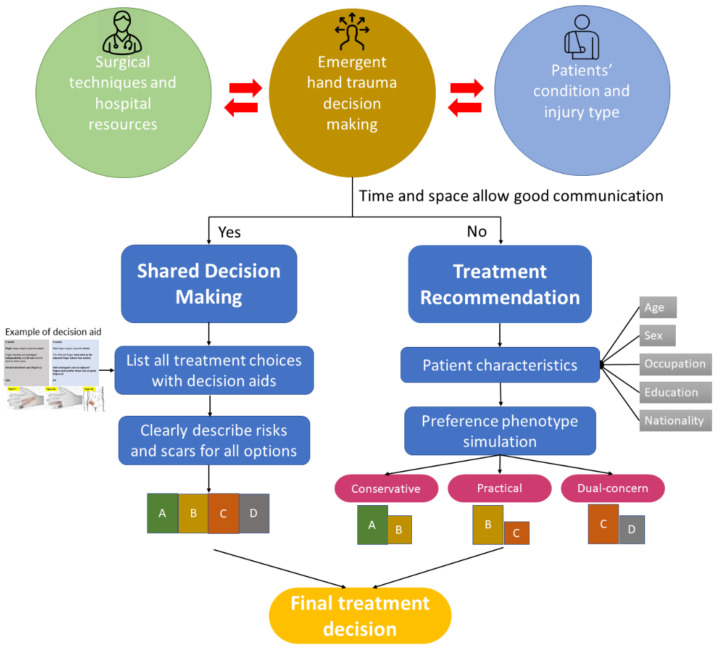
The proposed decision-making model incorporating patient preference.

**Table 1 ijerph-18-11081-t001:** Attribute and value levels of finger trauma reconstruction.

Attribute	Level 1	Level 2	Level 3
Recovery time	2 weeks	4 weeks	
Total procedures	One stage	Two stage	
Postoperative care methods	Two fingers stuck together (2 weeks)	Each finger cared for independently	
Postoperative scar condition	Dorsal hand linear scar	Square-shaped scar at the finger + groin scar	Forearm linear scar
Complication rate	1%	5%	10%

**Table 2 ijerph-18-11081-t002:** Patient demographics and characteristics.

Characteristics (N = 197)	*n* (%)
Sex	
Male	143 (72.6)
Female	54 (27.4)
Age	
20–29	38 (19.3)
30–39	43 (21.8)
40–49	39 (19.8)
50–59	43 (21.8)
60–69	24 (12.2)
>70	10 (5.1)
Foreign nationality	18 (9.1)
Vietnamese	15 (83.3)
Indonesian	1 (5.6)
Burmese	2 (11.1)
Education level	
Elementary school	12 (6.1)
Junior high school	39 (19.8)
High school	81 (41.1)
University (college)	55 (27.9)
Graduate school and above	10 (5.1)
Occupation	
Service industry	63 (32.0)
Worker and farmer	101 (51.3)
Student	6 (3.0)
No job or retired	27 (13.7)
Hand injury type	
Laceration	61 (31.0)
Only skin defect	34 (17.3)
Skin defect + tendon rupture	52 (26.4)
Skin defect + tendon rupture + fracture	31 (15.7)
Only fracture	19 (9.6)
Experience of skin graft or flap	39 (19.8)

**Table 3 ijerph-18-11081-t003:** Outcome of conjoint analysis of preference of reconstruction surgery in patients with hand trauma (*n* = 197).

Attributes and Levels	Average Utility (Range)	Attribute Importance %(SD)
Recovery time		9.55 (10.51)
2 weeks	12.61(−101.82 to 101.08)	
4 weeks	−12.61 (−101.08 to 101.82)	
Total procedures		13.86 (11.99)
Once	30.01(−85.20 to 161.28)	
Twice	−30.01(−161.28 to 85.20)	
Postoperative care methods		12.17 (10.74)
Connected two fingers	−26.55 (−118.88 to 69.16)	
Independent	26.55 (−69.16 to 118.88)	
Postoperative scar condition		21.55 (15.27)
Finger and groin scar	1.63 (−213.12 to 174.97)	
Finger and forearm scar	1.5 (−139.41 to 122.49)	
Dorsal hand linear scar	−3.24 (−191.49 to 279.07)	
Complication rates		42.87 (17.38)
1%	87.28 (−138.49 to 257.91)	
5%	17.68 (−137.57 to 255.95)	
10%	−104.97 (−229.34 to 105.27)	

**Table 4 ijerph-18-11081-t004:** Cluster analysis with the sociodemographic profiles of the participants.

Characteristics	Cluster 1 (*n* = 100) Conservative	Cluster 2 (*n* = 46) Practical	Cluster 3 (*n* = 51) Dual-Concern	Total	*p*-Value
Age	43.6 ± 14.3	46.2 ± 17.2	44.1 ± 14.5	44.1	0.606
Sex					0.330
Male	68 (68.0)	35 (76.1)	40 (78.4)	143	
Female	32 (32.0)	11 (23.9)	11 (21.6)	54	
Nationality					0.010
Local	95 (95.0)	43 (93.5)	41 (80.4)	179	
Foreign	5 (5.0)	3 (6.5)	10 (19.6)	18	
Education					0.041
High school/below	59 (59.0)	33 (71.7)	40 (78.4)	132	
College/above	41 (41.0)	13 (28.3)	11 (21.6)	65	
Occupation					0.086
Blue collar	45 (45.0)	24 (52.2)	32 (62.7)	101	
White collar	37 (37.0)	13 (28.3)	13 (25.5)	63	
Student	12 (12.0)	9 (19.6)	6 (11.8)	26	
None	6 (6.0)	0	0	6	
Experience					0.094
Graft or flap	15 (15.0)	14 (30.4)	10 (19.6)	39	
Others	85 (85.0)	32 (69.6)	41 (80.4)	158	

## Data Availability

The data presented in this study are available on request from the corresponding author. The data are not publicly available due to the concern of the patient privacy.

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
