# Peer review of "Incorporating Patient Preferences into a Decision-Making Model of Hand Trauma Reconstruction"

_ijerph, 2021, doi:10.3390/ijerph182111081_

Round 1

Reviewer 1 Report

Very useful study, particularly as it provides evidence for the use of shared decision making in hand trauma surgery.

Main comments:

  1. Statistics - There is no mention of the calculation of the sample size or power of the study. Without these, the results cannot be interpreted properly. Please update the manuscript to include this.
  2. Lines 248 & 254 -  The only scenario where communication would be impossible is if the patient lacks the mental capacity to provide informed consent.  Language should not be a barrier, and professional translators services (either in person or over the phone) must be sought to aid the consent process. Providing treatment without receiving informed consent is assault. Where the patient lacks mental capacity to make decisions, then the patient’s best interests in line with the legal framework of the country the operation is conducted in would guide the management strategy.

Some small comments:

Line 34 - “Preference Phenotype” this is not a common phrase to my (albeit limited) knowledge. Phenotype to me describes the physical properties of an organism. Can the authors expand on the meaning of this phrase more, or choose different words?

Line 187 - typo “looAking” should be “looking”

Line 276 - remove “fancy flap”, again line 281. Use a more professional term.

Reviewer 2 Report

Interesting study but references are missing. Please add and discuss the following references:

1: Sisti A, Oliver JD, Nisi G. Fenestrated adipofascial reverse flap for the
reconstruction of fingertip amputations. Microsurgery. 2019 Sep;39(6):575. doi:
10.1002/micr.30480. Epub 2019 Jul 3. PMID: 31268559.

2: Idone F, Sisti A, Tassinari J, Nisi G. Fenestrated Adipofascial Reverse Flap:
A Modified Technique for the Reconstruction of Fingertip Amputations. J Invest
Surg. 2017 Dec;30(6):353-358. doi: 10.1080/08941939.2016.1251667. Epub 2016 Nov
30. PMID: 27901645

3. Harbour PW, Malphrus E, Zimmerman RM, Giladi AM. Delayed Digit Replantation:
What is the Evidence? J Hand Surg Am. 2021 Aug 7:S0363-5023(21)00435-4. doi:
10.1016/j.jhsa.2021.07.007. Epub ahead of print. PMID: 34376294.

4. Samantaray SA, Oommen J, Thamunni CV, Kalathingal K, Koyappathody HM, Shet
SM, Joseph S, Pydi RV. Fingertip injury epidemiology: an Indian perspective. J
Plast Surg Hand Surg. 2021 Aug 8:1-5. doi: 10.1080/2000656X.2021.1962332. Epub
ahead of print. PMID: 34369266.

5. 1: Billington AR, Ogden BW, Le NK, King KS, Rotatori RM, Kim RL, Nydick J. A
17-Year Experience in Hand and Digit Replantation at an Academic Center. Plast
Reconstr Surg. 2021 Aug 16. doi: 10.1097/PRS.0000000000008314. Epub ahead of
print. PMID: 34398867.

The Abstract is not clear. I suggest to structure the Abstract in:

  Background   Aim of the study   Materials and Methods   Results   Conclusions  

Reviewer 3 Report

Overall, this is a good quality research paper with a clinical data set and solid analytical method. This manuscript is well written and addresses a critical topic. The literature review should be further discussed and the findings presented in a clearer manner. I would like to have seen a more detailed description of the data analysis procedures. Specifically, I think it would be helpful to provide a brief theoretical description of the conjoint analysis and explain why it is appropriate in this study. Moreover, this study is suggested for some grammatical errors to check carefully.
